# WORDSWORTH SCORES FOR ATTACKING CNNS AND LSTMS FOR TEXT CLASSIFICATION

## ABSTRACT

Black box attacks on traditional deep learning models trained for text classification target important words in a piece of text, in order to change model prediction. Current approaches towards highlighting important features are time consuming and require large number of model queries. We present a simple yet novel method to calculate word importance scores, based on model predictions on single words. These scores, which we call WordsWorth scores, need to be calculated only once for the training vocabulary. They can be used to speed up any attack method that requires word importance, with negligible loss of attack performance. We run experiments on a number of datasets trained on word-level CNNs and LSTMs, for sentiment analysis and topic classification and compare to state-of-the-art baselines. Our results show the effectiveness of our method in attacking these models with success rates that are close to the original baselines. We argue that global importance scores act as a very good proxy for word importance in a local context because words are a highly informative form of data. This aligns with the manner in which humans interpret language, with individual words having well-defined meaning and powerful connotations. We further show that these scores can be used as a debugging tool to interpret a trained model by highlighting relevant words for each class. Additionally, we demonstrate the effect of overtraining on word importance, compare the robustness of CNNs and LSTMs, and explain the transferability of adversarial examples across a CNN and an LSTM using these scores. We highlight the fact that neural networks make highly informative predictions on single words.

## 1 INTRODUCTION

Deep learning models are vulnerable to carefully crafted adversarial examples. The goal of such an attack is to fool a classifier into giving incorrect prediction while the perturbed input appears normal to human observers. The probelm is important from the point of view of robustness as well as interpretability. Thoroughly analyzing different kinds of vulnerabilities in neural networks would help us in creating robust models for deployment in the real world, in addition to throwing some light on the internal working of these models. In this work, we consider text classification, where finding important words in a body of text is the first step towards malicious modification.

For this problem, we propose a novel method for calculating word importance. After training a model, we calculate importance scores over the entire training vocabulary, word by word. We further use these importance scores for black box attacks and demonstrate that the attack success rate is comparable to the original methods, particularly for CNNs.

Since these scores are global and calculated over the training vocabulary, they can also be used as a tool to interpret a trained model. They provide a measure for comparing different architectures and models beyond training and validation accuracy. Over a single training dataset, we can compare a small CNN to a large CNN, a CNN to an LSTM, or the word importance distribution of one class against another, as we outline in our experiments section.

The motivation for our particular algorithm comes from the fact that in a piece of text, most of the time, words and phrases have a strong influence on their own. This gives us a rationale for evaluating a model on single words, in direct contrast to the leave-one-out technique, which involves deleting a word from a document and measuring its importance by the change in model prediction on this

modified input.

Further, we expect a well-trained network to treat a word approximately the same, irrespective of its location in the input, when surrounding words are removed. Thus a particular word can occur at any position in a document with 200 words and its importance will be roughly the same. We expect a well-trained model to exhibit this behaviour and our experiments confirm this.

In summary, our contributions are as follows:

- We propose a simple and efficient method for calculating word importance for attacking traditional deep learning models in the black box setting.

- We argue that these scores can act as a tool for model interpretation and outline a number of use cases in this context.

## 2 RELATED WORK

### 2.1 ADVERSARIAL ATTACKS ON NLP MODELS:

The idea of perturbation, whether random or malicious, is rather simple in the image domain, where salt and pepper noise can be enough to images to fool models. This kind of noise is hard for humans to detect. However, since text data is discrete, perturbations in text are difficult to quantify. Besides, people easily notice errors in computer-generated text. This places additional constraints for an NLP attack to be counted as successful, where a successful attack is one that forces the model to give an incorrect prediction while a human would make the correct prediction on the input.

We limit ourselves to text classification problems, using sentiment analysis and topic classification as examples. We only consider the attack scenarios in which specific words in the input are replaced by valid words from the dictionary. Thus we are not considering attacks in which extra information is appended to input data, or where word replacements purposefully introduce spelling errors. The former take an entirely different approach; the latter introduce errors and do not preserve semantics. In addition, training a neural network to be robust to spelling errors would stop these attacks. Further, we limit ourselves to black box attacks where the attacker has no information about model architectures and parameters.

### 2.2 FIND AND REPLACE ATTACKS ON TEXT CLASSIFICATION

Most attacks on text classification solve the problem in two parts; by locating important words in the input, and by finding suitable replacements for these words. We only consider attacks where substitutions are valid words picked from a dictionary, to avoid introducing grammatical errors, and ignore the case, for example, when spelling errors are introduced in important words.

#### 2.2.1 WHITE BOX ATTACKS

In the white-box setting, where an attacker has full knowledge of the model architecture, gradients serve as a good proxy for word importance. Gong et al. (2018) use gradient based methods to locate important words. Samanta & Mehta (2017) use gradients to calculate word importance, with linguistic constraints over substitution words. Lei et al. (2019) carry joint word and sentence attacks, by generating sentence paraphrases in the first stage, and resorting to greedy word substitutions if the first stage fails. Again, important words are located by the magnitude of the gradient of word embedding.

#### 2.2.2 BLACK BOX ATTACKS

In the black box scenario, where gradients are not available, saliency maps are calculated for words through different methods. Yang et al. (2018) provide a greedy algorithm which we will outline in detail in the next section.

Li et al. (2016) propose masking each feature with zero padding, using the decrease in the predicted probability as the score of the feature or word, and masking the top-k features as unknown. Alzantot et al. (2018) and Kuleshov et al. (2018) propose variations of genetic algorithms. Kuleshov et al. (2018) replace words one by one until the classifier is misdirected while observing a bound on the

number of perturbed features. They run each new iteration on the modified input. For substitution, they used post processed GloVe to find pool of suitable words. They also compute 'thought vectors' for sentences and ensure that these are preserved. Alzantot et al. (2018) select words by random sampling, where probability of each word being selected is proportional to the number of suitable neighbours for replacement. They use Google 1 billion words language model to ensure that replacements match the context provided by the rest of the input. Ren et al. (2019) propose a saliency-based greedy algorithm, calculated by deleting words during the search phase and select substitutions from WordNet. Another similar attack model is Jin et al. (2019), which has extra semantic similarity checking when searching adversarial examples, and calculates word importance by deleting words.

Zang et al. (2019) propose a particle swarm optimization algorithm for the search problem. Gao et al. (2018) define different scoring functions where they look at prediction before and after removing a particular word from a subset of input, and perform character level modifications in the second stage. Li et al. (2019) use the sentence probability directly but once again, when ranking words, they try masking words in a sentence.

A common thread among all search methods for black box attacks is erasure or omission, where the effect of a word is observed by comparing the classifier output probability for original input to that for input with this particular word removed or replaced by zero.

### 2.3 Interpretability in machine learning through erasure

Li et al. (2016) is a pioneering body of work in the domain of interpretability that highlights the importance of interpreting networks by erasing parts of various layers. This Leave-One-Out method is followed by most interpretation algorithms. For a particular word, they calculate importance score as the average of prediction difference due to erasing this word from all test examples. Feng et al. (2018) gradually remove unimportant input words so that only the important ones are left at the end. Barham & Feizi (2019) propose sparse projected gradient descent to generate adversarial examples to improve interpretability. Nguyen (2018) looks at different methods of local explanations for labels, which include LIME, random feature deletion and first derivative saliency. Kádár et al. (2017) measure salience of a word by removing it and noting the change in prediction. Jin et al. (2019) mention deleting a particular word to calculate its importance score. Ren et al. (2019) use word saliency which is the change in the classifier output if a word is set to unknown. Carter et al. (2018) find sufficient input subsets while calculating the feature importance by masking words. For calculating word score matrices, Xu & Du (2020) propose a method which involves masking words. We want to highlight the aspect that all the dominant techniques for interpretation use leave-one-out method for calculating word importance. WordsWorth scores provide a reliable way of calculating feature importance, as shown by attack success rates. Thus, they can be reliably used to interpret a model after it has been trained. When these scores show that a particular word is important or unimportant for predicting a particular class, we can be sure that this is how the model behaves.

## 3 Greedy algorithm for black box attacks

The greedy algorithm mentioned in Yang et al. (2018) consists of two steps: finding the most important words in a text, and finding the most distracting replacements for these words, with some constraint. For an attack where k features are allowed to be perturbed, the top k important words are picked first, and then replaced one by one. In the first step, greedy finds the most important words by calculating importance scores for each word in the input using leave-one-out technique. The score of a word is the difference in prediction probability for original input and for input with the word removed. The second step of the algorithm includes finding suitable replacement for these words. Throughout this paper we will use their greedy algorithm as a baseline for comparison, since it achieves the highest success rate among all black box methods (Hsieh et al., 2019).

Greedy uses the pretrained GloVe embeddings and limits the search in second step to within a pre-specified distance, to preserve semantics. However, it should be noted that GloVe embeddings do not always provide semantic preserving replacements, and a post-processed form of embeddings would work better, such as the ones used by Kuleshov et al. (2018). In our experiments, we use 50-dimensional GloVe embeddings to find replacements for important words. We limit our search to the ten nearest neighbours for each word.

# 4 WORDSWORTH SCORES FOR FEATURE IMPORTANCE

For determining importance of individual words in a text document, we propose WordsWorth scores, which are the prediction scores of each individual word in the vocabulary, from the trained classifier. Since the CNNs and LSTMs on which we experiment have a fixed input size(limited to 200 words throughout the experiments), for calculating these scores, the integer representation of the word (from the tokenizer) is appended with zeros and fed to the classifier. This is equivalent to evaluating the classifier on a piece of text where the text consists of a single word. The algorithm for greedy attack using WordsWorth scores is given below.

---

**Algorithm 1** Step 1: Calculate WW scores over $\mathcal{V}$, the training vocabulary

---

    **Input** $\mathcal{F}$, a trained CNN or LSTM
    **Input** p the number of classes in data
    **Input** d the size of classifier input
    **Input** $\mathcal{V} \in R^d$, the training vocabulary having m words
    **Output** $WW \in R^{d*p}$, Wordsworth scores over the training vocabulary
1: **for** $w \in \mathcal{V}$ **do**
2:     define $x = 0_0, 0_1, 0_2, ...0_{d-2}, w$
3:     $WW(w) = \mathcal{F}(x)$
4: **end for**

---

---

**Algorithm 2** Step 2: Greedily replace top k words to maximize incorrect class probability

---

    **Input** $\mathcal{F}$, a trained CNN or LSTM
    **Input** $X \in R^d$, text input to be modified
    **Input** $k$ the maximum number of features to be perturbed
    **Input** $D \in R^{m*10}$ the nearest neighbour dictionary with 10 neighbours for each training vocab word
    **Input** $WW \in R^{d*p}$, Wordsworth scores over the training vocabulary
    **Output** $X'$, the maliciously modified input with maximum k words modified
1:
2: Pick $i_1, i_2, .., i_k$ such that $WW(X_{i_1}) >= WW(X_{i_2}).. >= WW(X_{i_k})$
3: Initialize $X' = X$
4: **for** $j \in i_1, i_2, .., i_k$ **do**
5:     $w_j = X'_j, D_j = 10$ nearest neighbour of $w_j$
6:     **for** $w \in D_j$ **do**
7:
$$X'_w = \begin{cases} X'_j & \text{if } j \neq i \\ w & \text{if } j = i \end{cases}$$
8:     **end for**
9:     $X' = \arg_{X'_w} \max |\mathcal{F}(X'_w) - \mathcal{F}(X')|$
10: **end for**

---

# 5 EXPERIMENTS

Comparison with two other blackbox attacks: Here we present the performance of del_one (Li et al., 2016) and $greedy$ (Yang et al., 2018), along with their modified versions, where word importance has been computed through WordsWorth scores for the modified versions. We call the modified versions as del_one_ww and greedy_ww respectively. We also show the AUC scores for original data, named as original, to serve as a baseline.

## 5.1 SENTIMENT ANALYSIS: IMDB REVIEWS

### 5.1.1 DATASET AND MODEL ARCHITECTURE

We use the IMDB dataset (Maas et al., 2011), which consists of 25000 training reviews and 25000 test reviews of variable length. Each review in the training set has a positive/negative label attached

to it. Training vocabulary size is 5000 and we cut each review to 200 words max. We use a simple CNN as the starting point of our experiments, with 32 dimensional embedding layer, 50 filters and 100 units in a dense layer. The input layer uses word2vec embeddings that are learned during training. ReLU activation is used. The network is trained on 25000 training examples. We use the Adam optmizer with default learning rate of 0.001 and use early stopping. Validation accuracy is 88.78%.

We pick 300 examples at random from the test dataset and plot the ROC AUC values versus number of features perturbed for different algorithms. The results for CNN in figure 1(left) show that the modified versions of both algorithms have a performance that is comparable to that of the original versions. The distance between greedy and greedy_ww is larger than that between del_one and del_one_ww. This implies that if simply deleting words is the strategy, WordsWorth scores are almost as effective as manually deleting each word one by one and finding the one that contributes most to the model prediction.

### 5.1.2 RUNTIME COMPARISON WITH BASELINES

WordsWorth scores can be calculated once over the vocabulary learnt during training once the classifier has been trained. At test time, model evaluation can be replaced by a simple lookup. Thus, for a 5000 vocabulary size, WW score calculation takes 5000 model prediction. On the other hand, with 200 word reviews on average, the original baselines(greedy as well as del_one) need 5000 evaluations to locate important words just for 25 text examples, because they involve deleting each word in a review to calculate its importance only for this particular review. Thus, if a word appears in multiple reviews, its importance has to be calculated separately in the context of each review, for greedy and del_one.

If attacks are carried out in bulk, WordsWorth evaluations are essentially free after the first 25 reviews. This indicates a considerable slashing of computation time and resources. Additionally, WordsWorth score computations use a sparse input, which are more suitable for low-power platforms as compared to greedy and del_one.

### 5.1.3 DO GREEDY AND GREEDY_WW FIND THE SAME WORDS TO BE IMPORTANT?

During this experiment, when we compared the top ten words found by greedy and greedy_ww for each test example, 7.3 words were same on average. When we looked at the top 5 words, 3.5 were same on average. This strengthens the idea that both algorithms choose quite similar set of words for each instance.

### 5.1.4 LSTM

We repeat the experiment on an LSTM with 100 examples chosen at random and report the results in figure 1(right). Here, similar trends can be observed, with del_one_ww performing close to del_one and greedy_ww performing close to greedy. However, the difference here is larger as compared to CNN, which could be due to the LSTM learning a more robust representation.

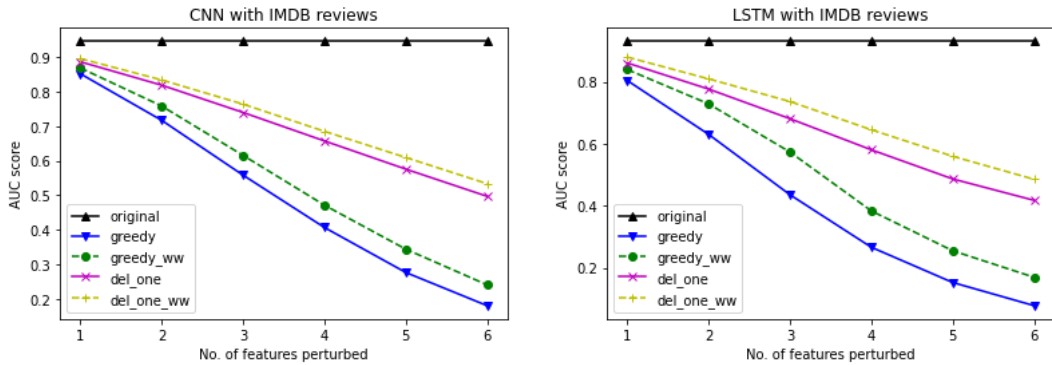

Figure 1: Left: AUC score vs words modified for IMDB Reviews with CNN Right: IMDB Review with LSTM

## 5.2 SENTIMENT ANALYSIS: YELP REVIEWS

The Yelp reviews dataset consists of positive and negative reviews. We train a CNN with 32 input units, 32 filters and 64 hidden units with Relu activation. We use 83200 training example and 15000 validation examples. The CNN has 89.96% train accuracy, 93.74% validation accuracy. We use the Adam optmizer with default learning rate of 0.001 and use early stopping. The input layer uses word2vec embeddings that are learned during training. We carry out attacks on 500 random test examples and report results in figure 2(left). Here we show the accuracy of the classifier on all 500 examples as the number of perturbed features increases. Here we have added replace_random and delete_random as two additional baselines. Replace_random replaces k features chosen at random, whereas delete_random deletes k random features.

## 5.3 TOPIC CLASSIFICATION: AG NEWS

The AG news dataset consists of news related to 4 categories. We train a CNN with 32 input units, 32 filters and 64 hidden units with ReLU activation. It has 96000 training example and 24000 validation examples. We train for 2 epochs with 96.4% train accuracy, 94.8% validation accuracy. We use the Adam optmizer with default learning rate of 0.001 and use early stopping. The input layer uses word2vec embeddings that are learned during training.
We carry out attacks on 500 random test examples and report results in figure 2(right). Here we show the accuracy of the classifier on all 500 examples as the number of perturbed features increases. The results on this multiclass dataset confirm that WW scores are a good proxy for local word importance. The attacks here are untargeted, with the objective being to minimize the correct class probability. Targeted attacks can also be similarly launched using these scores.

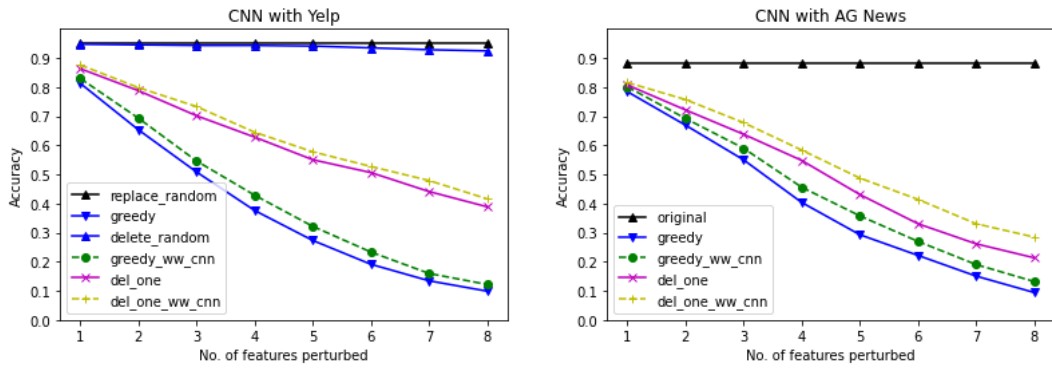

Figure 2: Left: Accuracy vs words modified for Yelp reviews with CNN Right: Accuracy vs words modified for AG News with CNN

## 5.4 ADDITIONAL EXPERIMENTS ON IMDB REVIEWS

In this section we describe a number of other experiments we ran on the IMDB reviews dataset.

### 5.4.1 IS GREEDY A LOWER BOUND FOR ATTACK SUCCESS?

We carried out further experiments by creating a new algorithm that evaluates greedy and greedy_ww for each test example and chooses the best result of both. If greedy and greedy_ww were finding different types of vulnerabilities, we would have expected the algorithm to perform better than both. In fact, the algorithm did no better than greedy, and thus greedy appears to be a bound for greedy_ww. Recall that in greedy, feature deletion is followed by feature insertion, so it does not follow directly that evaluation of input with a feature deleted should perform better than evaluation with everything except the feature deleted.
We hypothesize that the success of greedy attacks is partially explained by WordsWorth scores. Most of the times, greedy is just picking the words with the highest global importance and finding replacements. In some cases it optimizes further, which explains its improved performance over

greedy_ww. We would like to point out that a surprisingly high fraction of successful greedy attacks is explained by our single word scores. This suggests that most of the time, the impact of a word on the prediction is independent of its context.

### 5.4.2 THE CASE OF SMALL ARCHITECTURES

To test the algorithm with smaller networks, we train a very small CNN (8 dimensional embedding layer,8 filters and 16 units in a dense layer) and run our experiments on it. As with all other experiments, we use the Adam optmizer with default learning rate of 0.001 and use early stopping. The input layer uses word2vec embeddings that are learned during training. The test accuracy is the same as that for the larger CNN, but the model appears to be holding up better to greedy_ww attacks, as shown in figure 3(left). Compare the results to figure 1 (left), for a large CNN. This shows that text classification problems can be easily learned by relatively small CNNS, particularly when the number of classes is small.

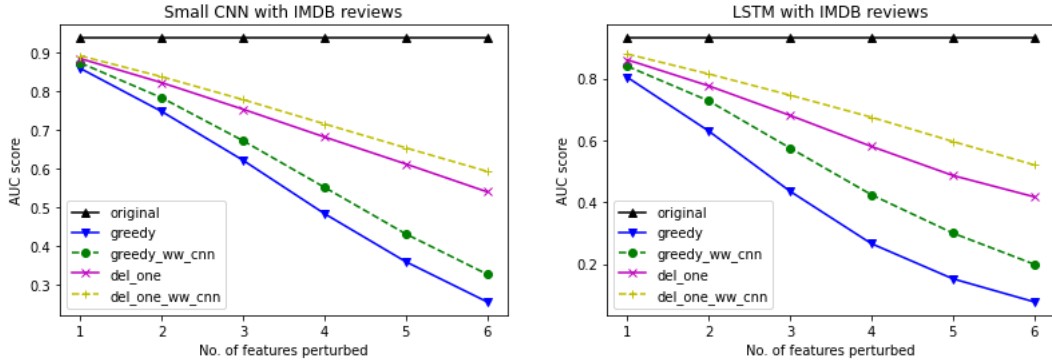

Figure 3: Left: AUC scores for IMDB reviews with a small CNN Right: AUC scores for LSTM, with WordsWorth scores from a CNN

The Pearson correlation between WordsWorth scores for this model and our main CNN is 0.83. The relatively poor performance of the bigger CNN could be due to overfitting. Since the task of binary classification is rather simple, the smaller network could be learning more robust and meaningful representations. However, contradictory hypotheses exist for images, such as Oscar Deniz & Bueno (2020).

Here, we highlight the fact that robustness to attacks, as well as score comparison, could be one interesting way to compare small vs big and deep vs shallow models.

### 5.4.3 TRANSFERABILITY AND SECURITY

The phenomenon of transferability is well documented in adversarial attacks on deep models, where adversarial examples generated for one trained model are often successful in attacking another model (Papernot et al., 2016). To demonstrate the phenomenon of transferability, we attack an LSTM with greedy and del_one, and with greedy_ww_cnn and del_one_ww_cnn where the WordsWorth scores have been calculated through a CNN, and the second step in adversarial search is evaluated directly on the LSTM. The correlation between the CNN and LSTM WW scores came out to be 0.88. Results of the attacks are shown in figure 3(right).

There is some drop in performance but still a noticeable degree of success. The close alignment of greedy and greedy_ww_cnn shows that the importance scores calculated through CNN are valid for LSTM too, even though directly using LSTM scores gives better performance. Compare this to figure 1(right) where LSTM was attacked with WW scores from the LSTM itself. We argue that this close, non-random alignment in figure 3 (right) explains the phenomenon of transferability in general. Features that are important for one architecture are important for another architecture too, when both models have been trained on the same dataset.

Our argument here has two parts: that scores from CNN and scores from LSTM have very high correlation(0.88), and scores from CNN can be used to attack LSTM(which is the transferability

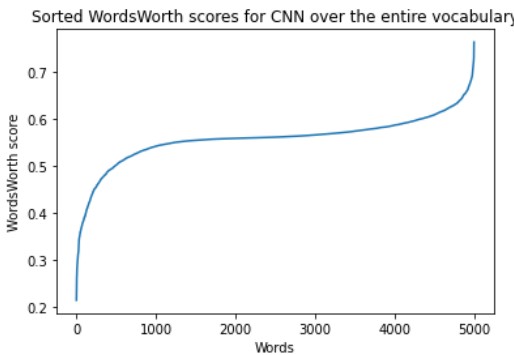

Figure 4: WordsWorth scores for CNN on training vocabulary on IMDB reviews

phenomenon) with a reasonable success rate , and that the former explains the latter.
Additionally, this highlights the aspect that for attacking a black box model, an adversary can train a small model locally and use it to highlight the vulnerable points of a piece of text, while using the black box model to find out substitutes, since the latter requires a much fewer number of model evaluations than the former.

# 6 INTERPRETING NEURAL NETWORKS THROUGH WORDSWORTH SCORES

In this section we show how to use WordsWorth scores for interpreting a model.

## 6.1 IMDB REVIEWS: LOCATING IMPORTANT WORDS

For the IMDB dataset, we computed the WordsWorth scores over our entire vocabulary (limited to 5000 top words) for the CNN as well as the LSTM. The top ten important words for the CNN are given in the table 2.In this manner, the model designer can directly find the top ranked words associated with each sentiment after training and examine errors in training. For the CNN, the mean WW score is 0.559 and standard deviation is 0.0530. We also include a snapshot of the scores for the entire vocabulary for the CNN in figure 4.

## 6.2 AG NEWS

We computed the WordsWorth scores over our entire vocabulary (limited to 20000 top words) for the trained CNN. The top ten important words for each category are given in the table 3. Here, 'martha' is the 9th most important word for category 'Business' and this potentially represents a generalization error, one that a model designer might want to investigate. Using score-based insights to actually improve generalization is a direction for future research.

### 6.2.1 AG NEWS WORD IMPORTANCE SCORE DISTRIBUTION

We also plot the scores for each class for AG News in figure 6 and 7. Different categories have different distributions associated with them. This is an interesting fact and could point to differences in writing style for each category, or to a difference in word distribution within each category.

## 6.3 OVERTRAINING AND WORD IMPORTANCE SCORES

We overtrain a CNN on the IMDB reviews dataset and calculate word importance scores over the training vocabulary after every epoch. A few snapshots are in figure 5. Stats related to training are shown in table 1. Overtraining has a noticeable effect on individual word scores, with score distribution becoming homogeneous throughout the epochs. Again, investigating how word scores evolve can serve as an additional tool for a model designer in addition to training and validation accuracy.

## 7 CONCLUSION

We consider the problem of quickly finding important words in a text to perturb in order to maximize the efficacy of black box attacks on deep NLP models, in the context of text classification. For this problem we present WordsWorth, a feature ranking algorithm that performs comparably well to the state of the art approaches, particularly when only a small number of feature perturbations are allowed, while being orders of magnitude faster by virtue of being essentially a lookup on training vocabulary.

We also use these scores as a tool model interpretation, compare different architectures and give a metric for evaluating performance beyond training and validation accuracy.

We also explain the phenomenon of transferability observed in text adversarial attacks and show that black box attacks can yield valuable information about the training dataset. All in all, we argue that text generated by humans is a highly compact and informative representation of data and the way neural networks interpret language aligns with human understanding.

Overall, we provide a method for evaluating importance in parallel with word erasure techniques. Combining the two techniques would yield even richer insights into the workings of models. Seen another way, WordsWorth attacks uncover a particular kind of vulnerability in deep models. Our work is the first step in designing a rule based algorithm to attack deep models that deal with text, and the next one would be to look at complex interactions. By aligning the performance of rule based algorithms with empirical methods currently popular in deep learning, we can improve our understanding of these otherwise blackbox models.

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

# A APPENDIX

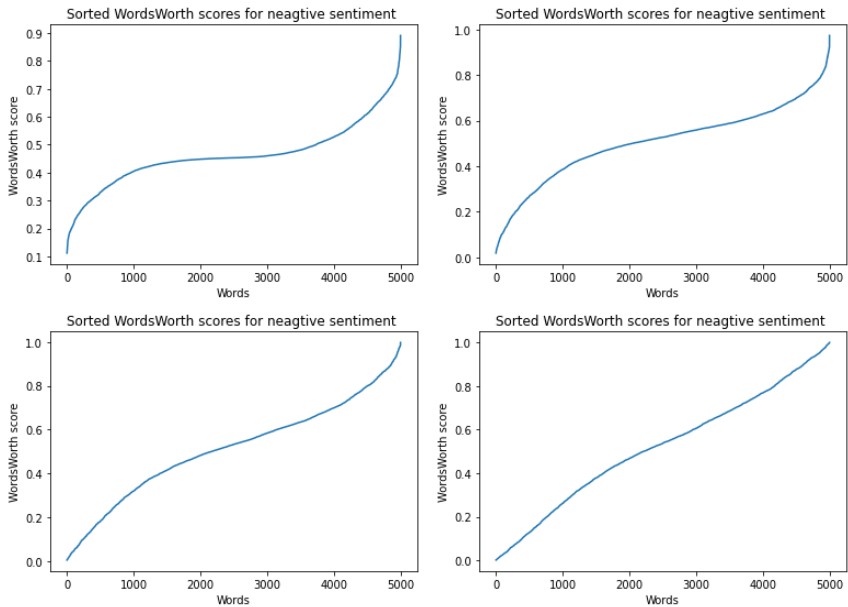

Figure 5: Overtraining CNN with IMDB Reviews. Top Left: After 4 epochs. Top Right: After 7 epochs. Bottom Left: After 10 epochs. Bottom Right: After 13 epochs.

Table 1: Stats for overtraining a CNN with IMDB Reviews

| epoch | Training loss | Training accuracy | Validation Loss | Validation Accuracy |
|-------|---------------|-------------------|-----------------|---------------------|
| 4 | 0.1678 | 0.9376 | 0.3187 | 0.8725 |
| 7 | 0.0208 | 0.9960 | 0.5919 | 0.8626 |
| 10 | 8.0602e-04 | 0.9999 | 0.8431 | 0.8625 |
| 13 | 7.6706e-05 | 1.0000 | 1.0780 | 0.8643 |

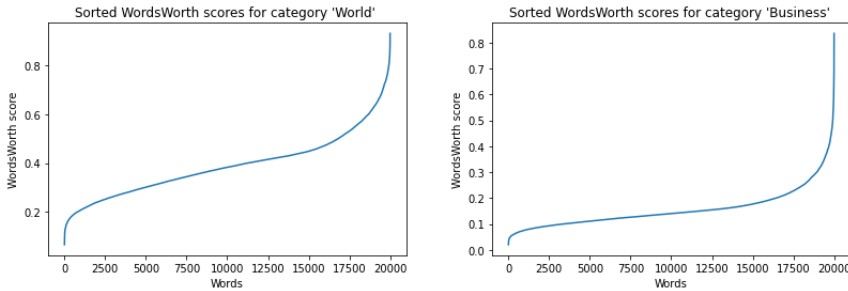

Figure 6: Left: scores for category World . Right: scores for category Business

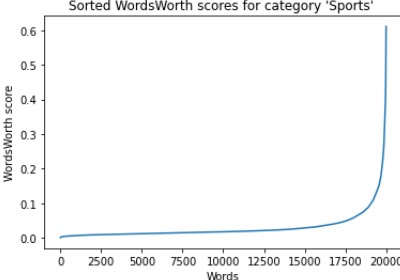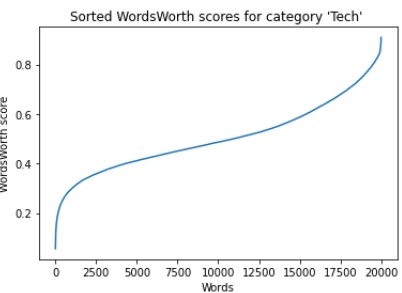

Figure 7: Left: Scores for category Sports. Right: Scores for category Tech

Table 2: Most important words for IMDB Reviews with CNN.

| Positive | Negative |
|---|---|
| perfect | waste |
| excellent | worst |
| rare | poorly |
| surprisingly | awful |
| refreshing | disappointing |
| wonderfully | forgettable |
| superb | fails |
| wonderful | disappointment |
| highly | pointless |
| outstanding | alright |

Table 3: Most important words for AG News for each class

| World | Sports | Business | Tech |
|---|---|---|---|
| dialogue | bcs | aspx | hypersonic |
| cayman | motorsports | bpd | voip |
| adultery | speedway | aeronautic | singel |
| grenade | nets | retreated | cybersecurity |
| takers | rockies | alitalia | processors |
| wangari | gators | mortgages | hacker |
| constitutional | quarterback | airlines | halo |
| sudanese | clippers | attendants | phonographic |
| israel | knicks | martha | healthday |
| militant | motorsport | opec | browser |

