# OpenReview forum: "WordsWorth Scores for Attacking CNNs and LSTMs for Text Classification"
_ICLR.cc/2021/Conference — Reject_

### Official Review · AnonReviewer2 · 2020-10-25
**An interesting idea but a paper clearly not ready for publication**

**Rating:** 4
**Confidence:** 4

**Review:**

This paper proposes a new and simple way to determine word importance for black box adversarial attacks on text classification models. Instead of using example-specific measures of importance like recent work (typically expensive to compute), the authors propose to feed individual words from the vocabulary to a trained model and use the model confidences to get global, class-specific importance scores.

While being an interesting paper, at a high level I am concerned about several points:
- The paper is unpolished and at times hard to follow. I do not consider it ready for publication at this stage.
- There are many easy to implement baselines (feature selection has a rich history) that would have been very interesting to study. Are the WW scores capturing anything that simpler statistical methods do not?
- Many implementation details are lacking, which could be an issue for reproducibility. For example, the CNN model details are unclear (number of layers, filter sizes, embedding initialization, etc.). Learning rates and other hyperparameters are not mentioned.
- Why is the vocabulary size only 5000? Why are experiments run on only 500 examples, and are these examples selected?
- The claim of comparable performance seems slightly exaggerated, as the WW scores perform consistently worse than the baselines. In some cases, the difference seems around 0.1 / 0.15 absolute AUC. Also, providing raw scores (either all scores in Appendix or a subset of the most interesting case in the text) would help readers to quantify these differences.
- Some of the claims seem overly broad. For determining, say, grammaticality, I would expect greedy to vastly outperform greedy_ww.
- It is unclear to me how WW scores help with network interpretation. Again I would expect WW scores to correlate with a number of statistical correlation measures.

There are many formatting issues, with figure numbers, references, equations, etc.

---

> ### Author Response · Authors · 2020-11-24
> **Response to the points raised in the review**
>
> Thank you for your review. Here is our response to all the points raised in the review:
>
> 1. We have updated the paper for improved presentation.
>
> 2. When traditional deep learning models(CNNs and LSTMs) are used, leave-one-out or some variant of it is the only technique that is used for feature importance, as outlined in the literature for attacks/interpretation.
>
> 3. We have added the learning rate, learning algorithm and embedding initialization method to the current revision. The architecture for all the experiments has been mentioned in the relevant sections.
>
> 4. Training vocabulary size of 5000 was a randomly chosen parameter, as was the test subset to be attacked.
>
> 5. In all experiments, the attack success rate for the original algorithms(greedy and delete one) is always better. However our technique is, on the surface, a very rough measure of actual word importance. We are claiming that in a paragraph of 200 words, the importance of any word can be determined approximately by removing 199 words and evaluating a classifier on just this single word. The decrease in success varies across architectures too(CNNs perform worse than LSTMs). Also notice that most of the times, the difference in performance is less than what would have been achieved with the original technique by perturbing one additional feature/word. However, we have modified the wording in the abstract to highlight the fact that there is some decrease in attack success.
>
> 6. This is a valid point, and it would be interesting to see how this kind of approach works on other NLP tasks. However, we have limited our analysis to text classification(sentiment analysis and topic classification). The technique is most suitable for scenarios where  a classifier learns to associate words with one label from a subset of labels. To determine grammaticality, context is extremely important, and our technique completely ignores context, to sacrifice accuracy for efficiency.
>
> 7. For interpreting a trained model, we show how to find important words for each class, provide lists in the appendix, and highlight that our model has learned that 'martha' is the 9th most important word for category 'Business' in the AG News dataset, which could potentially be a mistake we do not want our model to make.
> We also show that these scores explain transferability between a CNN and an LSTM. Our argument here has two parts: that scores from CNN and scores from LSTM have very high correlation(~0.88), and scores from CNN can be used to attack LSTM(which is the transferability phenomenon), and that the former explains the success of the latter.

---

### Official Review · AnonReviewer3 · 2020-10-28
**This paper provides a simple approach, but this paper lacks completeness.**

**Rating:** 3
**Confidence:** 3

**Review:**

Summary:
This paper proposes WordsWorth score (WW score), a score to represent the importance of the word obtained from the trained model. Then, the score is applied to the greedy attack proposed by (Yang et al., 2018). In detail, the greedy attack first tries to search for the most important $k$ words in a text, and then it searches for values to replace the selected $k$ words. This paper uses the WW score to select the $k$ words in the first step.

Strong points
+ A simple but effective approach to utilize for the greedy attack

Concerns:
- The main concern of this paper is that a minor contribution to the current knowledge. Despite the paper stating that this paper is based on the greedy attack (Yang et al., 2018), the contribution of this paper is limited to calculate the word score from the trained classifier and applied it to the greedy attack.

- Another concern about the paper is lack of rigorous experimentation to study the usefulness of the proposed method. This paper does not compare with other score-based approaches. That is, it was not even compared to the tf-idf based score approach.

- The writing should be largely improved. Section 4 is the main part of this paper. In Step 1, this paper represents a word as $d$-dimensional vector. Why does this paper append the zeros in front of the word representation? Does it mean the one-hot vector? If not, some studies or discussions about this representation should be included. In Step 2, Equations are hard to follow, and some are incorrectly written (e.g. case equation and definition of the D’).

- On the same note, readability and completeness of this paper do not meet the standard of conference. The reviewer suggests the authors to review the paper several times before submission.

- This paper states that $F$ is the trained classifier. However, there are no explanations on how to train or what kinds of classifiers were used.

- In Section 5.1.2, this paper states that it covers 5000 vocabularies freely after 25 reviews are processed because reviews are written in 200 words on average. It is definitely incorrect. Because all words in the review are not unique, it takes a longer timer to cache all words in the 5000 vocabularies. Although the proposed method can speed up by looking up the cached score, the performance of the proposed method is lower than the one of the original greedy approach. Some ablation studies or discussions about the relationship between speed and performance would have been useful to understand this.

- Some parameters or data are heuristically selected such as selecting 10 nearest neighbors in step 2, picking 300 examples from test data in IMDB review experiments, and so on. Some form of ablation studies about the parameters would provide appropriateness to readers.

- Furthermore, it would be better to show examples of successful attacks with WW scores.

- In experiments, the AUC score is used for IMDB evaluation and the accuracy is used for Yelp and AG news. Are there any reasons to use different evaluation measures?

Minor comments:
1. It would be better to write some constants such as $k$ in Section 3 as an italic character.
2. It will help readers if the authors can explicitly specify which side of the figure, left or right, is explained when the authors are referring to the figure in text.

Some typos:
1. In Section 5.1.1, greedyww -> greedy\_ww
2. In Section 5.4.2, figure ?? -> Figure 1
3. In Section 5.4.1, $greedy_ww$ -> greedy\_ww
4. In Section 5.4.3, figure ??
5. In Section 6.1, CNN 4 -> CNN at Figure 4
...

---

> ### Author Response · Authors · 2020-11-24
> **Some clarifications**
>
> Thank you for your review. Here is our response to the questions raised in the review:
>
> 1. Our contribution lies in highlighting that the predictions on a single word, while deleting the rest of the input, act as a surprisingly good proxy of feature importance. This is directly opposite to all the current approaches to measuring word importance, where a particular word is deleted from input sample and the change in prediction is used as importance score. This leave-one-out(LOO) method is the dominant technique in black box attacks as well as interpretation techniques(detailed literature review mentioned in the paper).
>
> 2. For feature importance, we agree that we have only compared with the leave-one-out method, and there are other interesting approaches. However, we limit our analysis to traditional deep learning models(CNNs and LSTMs) and for these models, LOO(or some variant of it) is the only black box technique that is used for calculating feature importance.  It would indeed be interesting to compare to frequency based measures, but we think the benchmark then would not remain fair, as our technique uses model predictions which are more informative than a simple frequency-based measure. Another point regarding experimentation is that we conduct experiments on three datasets(two binary, one multi-class) and different architectures.
>
> 3. Indeed the writing for this section is poor and we apologize for this, it has been improved in the updated version. Zeros are appended to a word to calculate its score because our models(CNN/LSTM) operate on fixed-width input, 200 words in our case. So to get inference on one word we construct an artificial input which consists of a single word.
>
> 4. Particular comments about writing, provided in the reviews, have been quite helpful, thank you for the review!
>
> 5. We conduct all experiments with CNNs and LSTMs. The architecture for every experiment has been mentioned. We have added learning rate and optimizer details in the updated version. Early stopping was used to recognize a good stopping point.
>
> 6. Indeed 5000 words(training vocabulary size in our experiments) would not be covered by just 25 reviews. We are claiming that the LOO approach would require 5000 model evaluations for attacking 25 reviews having 200 words each, because this technique measures importance by deleting individual words and thus analysis has to be repeated for each word when it appears in a new input sample(a different movie review, for example). On the other hand, our technique requires 5000 model queries initially, but nothing further.
>
> 7. These parameters were chosen at random. The test examples we use throughout our experiments are a random subset of test data. The 10 neighbour choice was borrowed from  (Yang et al., 2018).
>
> 8. Our main result is the close alignement between success rates of our techniques and traditional approaches. However, some examples would have been interesting to add, which we have missed.
>
> 9. AG news is a multiclass dataset(4 classes in all) so we used accuracy for reporting its result. For the other two binary datasets, we show accuracy for Yelp and AUC scores for IMDB because we think that AUC is an interesting metric for attacks. It incorporates the changes in prediction even when an attack isn't successful but the confidence score of the classifier decreases nonetheless.
>
> Overall, we have described all the components of our experiments, and verified our results on multiple datasets.We have compared to two baseline attack algorithms, greedy and delete one.
> Additionally, we show the use of these scores to gain some insight into the working of a trained model, such as finding important words for a class and the word importance distribution difference between different classes . We also show that the high correlation of these global scores from a CNN and an LSTM explains the transferability of adversarial attacks.

---

### Official Review · AnonReviewer1 · 2020-11-01
**The paper should be improved before submission**

**Rating:** 2
**Confidence:** 4

**Review:**

The paper is poorly written. This is especially the case for the
proposed algorithm (the core contribution). This section is very
difficult to understand, and notations are awkward. Everything is a
bit messy. However, the experimental results are quite well presented
(to be compared with the beginning of the paper).

---

> ### Author Response · Authors · 2020-11-24
> **Improved version uploaded**
>
> Thank you for your review. An updated version has been uploaded, with particular focus on the algorithm.

---

### Decision · Program_Chairs · 2021-01-07
**Final Decision**

**Decision:**

Reject

**Comment:**

The authors propose a method for attacking neural NLP models based on individual word importance ("WordsWorth" scores).  This is an interesting, timely topic and there may be some interesting ideas here, but at present the paper suffers from poor presentation which makes it difficult to discern the contribution. Presentation issues aside, it seems that the experimental setup is missing key baselines (an issue not sufficiently addressed by the author response).